# Projective Invariant Triad Indexing for Integrated Lunar Terrain Relative Navigation

Hyunsung Kim    Sunwoo Moon    Seokju Lee[†]

Korea Institute of Energy Technology (KENTECH)

{kimharry0610, m04041717, slee}@kentech.ac.kr

## Abstract

*Robust lunar terrain relative navigation (TRN) requires bridging the severe modality gap between offline heterogeneous orbital payloads and runtime monocular descent imagery. To seamlessly integrate these disparate views, we construct a 3D geometry-aware projective invariant triad index. By lifting 2D geo-referenced crater rim ellipse parameters into 3D disk quadric primitives, we extract appearance-independent geometric invariants that are independent of illumination and tolerant to viewpoint changes at the representation level. To evaluate how these properties propagate to trajectory-level performance, we tightly couple this index with a visual odometry (VO) module via an extended Kalman filter (EKF). Experiments using a QuickMap-based lunar south pole simulation systematically map the representation's operational envelope across observation noise, illumination, and off-nadir angles. Results show that within this envelope, sparse absolute geometric fixes suppress accumulated VO drift in mean position error by over 88%, demonstrating an integrated framework for continuous downstream lunar descent navigation.*

## 1. Introduction

During lunar powered descent and landing, the navigation system must maintain continuous and reliable onboard state estimation [14] throughout a time-critical flight regime in which guidance and control are tightly coupled to the navigation solution [27, 42]. However, in the cislunar environment, GNSS does not provide a continuously available, high-integrity absolute navigation reference for descent [1, 10, 30]. Lunar descent navigation is therefore typically solved by coupling two mutually complementary approaches. The first is relative navigation based on visual odometry (VO) [38], which provides high-rate, frame-to-frame relative motion estimates. However, monocular VO inevitably accumulates drift over time and suffers from

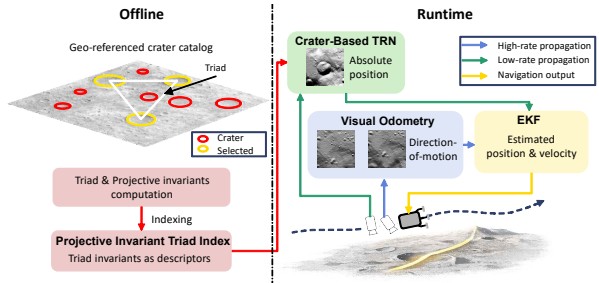

Figure 1. **Overview of the proposed integrated lunar TRN pipeline.** (Offline) A geo-referenced crater catalog is converted into disk quadric representations of 3D crater rims, grouped into triads, and indexed by their projective invariants to form a compact projective invariant triad index. (Runtime) A monocular VO module provides high-rate direction-of-motion estimates from descent imagery, while the crater-based TRN module returns absolute position fixes by matching observed crater triads against the precomputed index. Both measurement streams are fused in an extended Kalman filter (EKF), in which VO drives high-rate state propagation and crater-based localization corrects accumulated drift by resetting the estimated position.

scale ambiguity [19]. To correct this, map-based terrain relative navigation (TRN) [15, 16] is fundamentally required, providing absolute position fixes by matching onboard observations against a reference map or crater index [4]. In this tightly coupled architecture, VO ensures continuous state propagation while TRN acts as an absolute anchor. Nevertheless, if the sparse TRN updates fail under extreme variations in illumination or viewing geometry [4, 34], the entire state estimation collapses. Therefore, constructing a highly robust TRN representation is the critical bottleneck for maintaining this integrated navigation pipeline.

Constructing such a robust TRN representation is fundamentally challenged by the modality gap between heterogeneous observation platforms. Offline crater catalogs (raw data of craters, *e.g.*, the Robbins crater catalog [35]) are composite, macro-scale maps derived from heterogeneous orbital payloads, including both optical imagery and altimetry data. In contrast, the data fed to the crater-based

---

[†]Corresponding author.

TRN module consists of dynamic monocular images captured during descent, which suffer from perspective foreshortening and rapid illumination changes. Recent work by Christian *et al*. [4] establish a mathematical foundation for crater identification, demonstrating that crater rims modeled as 3D disk quadrics admit projective invariants for non-coplanar triads. Since these invariants are geometric quantities computable without appearance information, they offer a theoretical pathway to match heterogeneous views. However, their methodology focuses heavily on static, single-frame position recovery for the lost-in-space problem. It remains unexplored how these purely geometric properties behave under continuous dynamic trajectories, and whether they can seamlessly bridge the modality gap to reliably suppress VO drift in a downstream, tightly coupled navigation pipeline.

To address these gaps, we propose the construction of a projective invariant triad index (Fig. 1), a structure that maps configurations of non-coplanar crater triads to their corresponding projective-invariant 3-element descriptors. Unlike traditional image-based databases or conventional data indices, this projective invariant triad index is populated by lifting 2D geo-referenced parameters from orbital sources [35] into 3D disk quadric primitives, thereby creating a representation that is inherently independent from observer-specific appearance. By extracting geometry-aware descriptors from these 3D primitives, we provide a unified mathematical space that seamlessly integrates heterogeneous lunar observations across the modalities. To observe the inheritance of these geometric properties at scale, we integrate the proposed index with a monocular VO module [5] through an extended Kalman filter (EKF). In this configuration, the integrated pipeline serves as a systematic instrument to map the operational envelope of the 3D representation, explicitly characterizing its navigation performance under varying observation noise, illumination, and viewing geometry throughout a continuous descent trajectory .

In summary, our contributions are as follows:

◇ We construct a 3D geometry-aware projective invariant triad index that seamlessly integrates heterogeneous lunar observations. By lifting multi-sensor 2D orbital data into 3D geometric primitives, we bridge the gap with monocular descent imagery, making illumination invariance and bounded viewpoint tolerance inherent properties of the representation.

◇ We establish the structural necessity of coupling this crater-based TRN with a VO module. By fusing them through an EKF, we demonstrate how sparse, robust absolute geometric fixes complement the continuous high-rate state propagation of VO by resolving scale ambiguity and accumulated drift over a continuous dynamic trajectory.

◇ We construct a controlled, QuickMap-based simulation environment over the lunar south polar region to systematically map the operational envelope of our integrated pipeline. We explicitly characterize how the 3D geometric properties of the index propagate into trajectory-level robustness against variations in observation noise, illumination, and off-nadir viewing angle.

## 2. Related Work

### 2.1. Vision-Based Navigation and Terrain Relative Navigation for Lunar Landing

**VO for lunar landing.** VO estimates the relative motion from sequential descent images [38]. Representative approaches include feature-based visual odometry [38], semi-direct [11] and direct methods [9], and visual-inertial odometry [31] that combines image measurements with inertial sensing for relative motion estimation. In lunar landings, VO is primarily used as a relative navigation component [4, 28] that provides high-rate relative motion information during descent [28] rather than as a standalone absolute localization method [4].

**TRN for lunar landing.** The TRN estimates the state of the lander from observations of the terrain onboard during descent. In prior work on lunar TRN, representative TRN methods include crater detection [6–8] and crater-based matching to reference maps or landmark databases [6, 24], map-based descent localization [14, 15] using lunar reference maps such as orbital imagery and DEMs [15, 20, 34], and image-based terrain-relative measurements without an explicit map [5].

### 2.2. Projective-Invariant Crater Matching and Index Construction

**Projective-invariant crater matching.** Projective-invariant crater matching starts from conic invariants and invariant-based recognition. Early work by Forsyth *et al*. [12] introduce invariant descriptors for 3D object recognition and pose. Quan [32] shows that a pair of non-coplanar conics in space admits algebraic and geometric invariants, and Quan and Veillon [33] derive joint invariants for a triplet of coplanar conics and analyze their stability and discriminating power for object recognition. In crater-based TRN, the crater rims are modeled as image conics under perspective projection, and the crater triplets provide a natural unit for invariant indexing. Early lunar landing work focuses on detecting craters in an image and identifying them in a crater reference catalog [6], and later systems [24] match detected craters to a crater reference catalog for absolute navigation during descent. Christian *et al*. [4] establish an invariant-based crater matching framework. However, this foundational work is

strictly limited to static single-frame position recovery for the lost-in-space problem. It remains unexplored how these purely geometric descriptors behave across continuous dynamic descents or integrate into downstream navigation pipelines.

**Projective invariant triad index construction.** Beyond raw crater inventories, prior studies have already built precomputed matching indices for crater identification. Hanak *et al*. [13] match detected craters to entries in the USGS lunar crater catalog via non-dimensional crater triangle parameters. Kim *et al*. [18] propose a crater-matching method using projective invariant features derived from coplanar points and a simple voting algorithm. Park *et al*. [29] then present a crater-triangle matching algorithm for planetary landing navigation based on invariant crater-pattern descriptors. Christian *et al*. [4] later give the clearest large-scale formulation of this idea: crater triads are generated over a multi-scale HEALPix hierarchy, invariant descriptors are computed for each valid triad, and the resulting index is stored in efficient searchable structures such as $k$-d tree. Xu *et al*. [43] subsequently reduce the index size by constructing a crater pair database that stores serial numbers of crater pairs together with projective invariants and supports search-based binary retrieval. More recently, Chng *et al*. [3] argue that descriptor indices defined over crater tuples scale poorly with catalog size and remain sensitive to noise and outliers, and therefore proposed a descriptor-less alternative. Accordingly, the central open issue is not the absence of crater-index construction itself, but rather how to design a geometrically rigorous, noise-robust, and globally scalable matching index for lost-in-space crater identification.

## 3. Integrated Navigation Pipeline

### 3.1. System Overview

Fig. 2 illustrates our integrated navigation pipeline. The EKF fuses high-rate direction-of-motion estimates from a map-free monocular VO [5] with sparse absolute position fixes returned by matching observed crater triads against our offline projective invariant triad index.

### 3.2. Relative Motion Estimation

For high-rate state propagation between crater-based TRN updates, we adopt the map-free VO framework of Christian *et al*. [5]. The framework estimates the spacecraft's direction of motion from sequential monocular images without requiring an onboard terrain map or knowledge of observed landmark locations.

Local image features are extracted using standard detectors (ORB [37] or SIFT [23]), matched across consecutive descent images, and verified via RANSAC. Given the matched pixel coordinates and the relative camera attitude from inertial sensors (IMU), the epipolar constraint reduces to a linear system in the unit direction of motion $\mathbf{s}'_{(C_k)_k}$, whose unbiased maximum likelihood solution is obtained by minimizing the weighted Sampson distance [5]. The output is the unit vector $\tilde{\mathbf{s}}'_{(C_k)_k}$ with rank-2 covariance $\mathbf{R}_{\mathbf{s}'_k}$.

According to our experiments, ORB yields comparable angular error to SIFT ($0.386°$ vs. $0.367°$) at half the runtime per frame. Since the VO module must maintain high-rate propagation on embedded platforms, we select ORB as the feature detector.

This VO framework provides two properties essential to the integrated pipeline: continuous operation without an onboard map, and a unit-vector output whose inherent scale ambiguity motivates the fusion with map-referenced absolute localization.

### 3.3. 3D Crater Geometry-based TRN

The crater-based TRN module is the component through which the 3D geometric representation enters the navigation pipeline. Its role is to return absolute position fixes by matching observed crater rims against a precomputed catalog whose entries are 3D geometric primitives, not image templates. We build on the crater identification methodology of Christian *et al*. [4], and organize this section with the three properties of the representation that are directly inherited by the downstream navigation task: its 3D geometric nature, its non-coplanar structure dictated by lunar surface curvature, and the absence of appearance information in its matching primitives.

**3D crater representation.** The rim of the crater $i$ lies on a plane $P_i$ determined by local surface topography. Because the Moon is a curved surface, the planes supporting distinct crater rims are non-coplanar, which means that the tangent planes at two distinct lunar surface points never coincide.

The 3D conic is represented by its disk quadric $\mathbf{Q}_i^*$, a $4 \times 4$ symmetric rank-3 matrix that encodes both the shape of the elliptical rim and the orientation of its supporting plane [4]. The disk quadric is constructed from the crater catalog entry (center latitude/longitude, semi-axes $a$, $b$, orientation $\psi$) and a local East-North-Up frame via

$$\mathbf{Q}_i^* \propto \begin{bmatrix} \mathbf{H}_{Mi} \\ \mathbf{k}^\mathsf{T} \end{bmatrix} \mathbf{C}_i^* \begin{bmatrix} \mathbf{H}_{Mi} \\ \mathbf{k}^\mathsf{T} \end{bmatrix}^\mathsf{T} \qquad (1)$$

where $\mathbf{C}_i^*$ is the $3 \times 3$ conic envelope of the rim within its local plane, $\mathbf{H}_{Mi}$ maps local 2D coordinates to the global frame, and $\mathbf{k}^\mathsf{T}$ is a selective vector $[0\ 0\ 1]$.

This construction is a lifting of a 2D catalog entry into a 3D geometric primitive. The rim parameters in the Robbins crater catalog [35] describe a 2D ellipse in the local tangent plane. The disk quadric $\mathbf{Q}_i^*$ embeds this ellipse

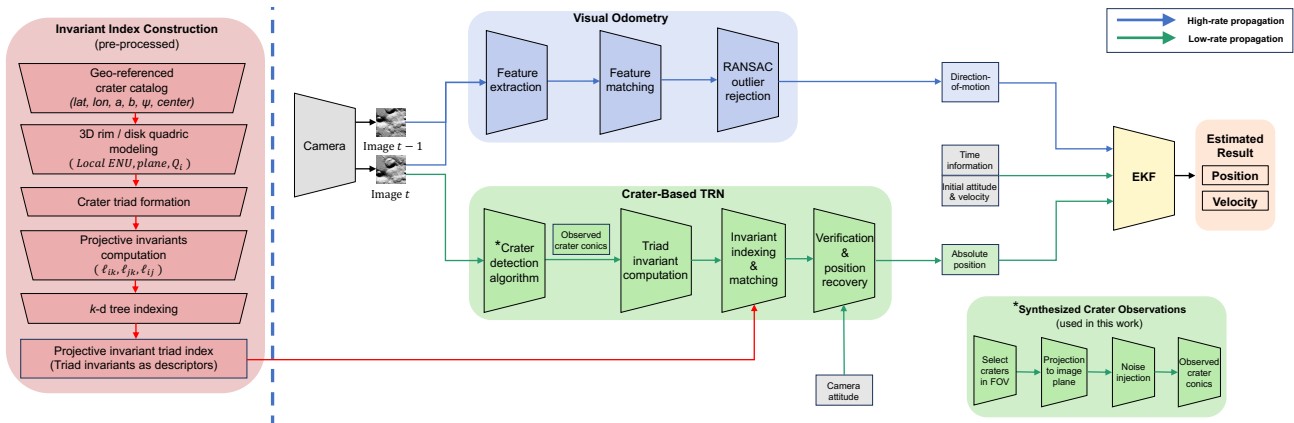

Figure 2. **Integrated TRN pipeline overview.** The projective invariant triad index (left) is constructed offline from the Robbins crater catalog [35], which provides geo-referenced crater rim parameters (center latitude/longitude, semi-axes, and orientation). Each cataloged rim is modeled as disk quadrics $\mathbf{Q}_i^*$, computing triad projective invariants, and indexing them in a $k$-d tree. At runtime (right), the VO module provides high-rate direction-of-motion estimates, the crater-based TRN module matches observed crater conics against the catalog for absolute position fixes, and both are fused in an EKF. In the present evaluation, the crater detection algorithm (CDA, marked with *) is bypassed. The crater observations are synthesized by projecting cataloged disk quadrics into the image plane (retaining only rims inside the FOV) and injecting pixel-level Gaussian noise into the resulting ellipse parameters. This protocol isolates the matching and pose recovery stages from upstream detection errors, following [4].

back into 3D space together with its supporting plane, producing a geometric object that is well-defined in the lunar body-fixed frame independently of any observer. The downstream matching primitives are derived from this 3D object, not from the 2D catalog entry alone.

**Projection to 2D conics.** At runtime, the disk quadric is projected onto the image plane as a conic envelope under perspective projection through a pinhole camera with projection matrix $\mathbf{P}$, given by

$$\mathbf{A}_i^* \propto \mathbf{P}\mathbf{Q}_i^*\mathbf{P}^\mathsf{T} \qquad (2)$$

from which the conic locus $\mathbf{A}_i$ is obtained by adjugation. This projection step is the only stage at which the observer and the representation interact. The projection matrix $\mathbf{P}$ depends on the camera pose, but the projective relationships among the multiple craters observed in a single image are preserved. If $\mathbf{Q}_i^*$ and $\mathbf{Q}_j^*$ satisfy a projective relation in 3D, their image conics $\mathbf{A}_i$ and $\mathbf{A}_j$ satisfy the corresponding relation in 2D. This is the structural reason why invariants computed from the image conics can recover information about the underlying 3D geometric configuration.

**Projective invariants of crater triads.** Christian *et al.* [4] show that $d \geq 3$ conics constrained to lie on a common non-degenerate quadric surface (an excellent model for the Moon) have $3d - 6$ algebraically independent projective invariants. For a triad ($d = 3$), this yields three independent invariants $J_i, J_j, J_k$, each computed as a Cayley-Klein distance between a pair of lines derived from the pencil of the

corresponding image conics:

$$J_i = \mathrm{acosh}\left\{ \frac{\|\boldsymbol{\ell}_{ij}^\mathsf{T}\mathbf{A}_i^*\boldsymbol{\ell}_{ik}\|}{\sqrt{\left(\boldsymbol{\ell}_{ij}^\mathsf{T}\mathbf{A}_i^*\boldsymbol{\ell}_{ij}\right)\left(\boldsymbol{\ell}_{ik}^\mathsf{T}\mathbf{A}_i^*\boldsymbol{\ell}_{ik}\right)}} \right\} \qquad (3)$$

where $\boldsymbol{\ell}_{ij}$ is the image-plane projection of the 3D line formed by the intersection of the planes $P_i$ and $P_j$, recoverable from only the observed image conics $\mathbf{A}_i$ and $\mathbf{A}_j$ via eigendecomposition of a degenerate pencil.

Two aspects of this construction are central to the present work. First, the non-coplanar structure of the lunar surface is necessary for the invariants to exist in their three-independent form. The line $\boldsymbol{\ell}_{ij}$ is defined as the intersection of the two supporting planes $P_i$ and $P_j$, and this intersection can only be defined because $P_i \neq P_j$. If the triad were coplanar, the supporting planes would coincide and the construction would degenerate. The lunar curvature discussed earlier guarantees that this degeneration does not arise for any triad of distinct craters. Second, invariants $J_i, J_j, J_k$ are functions of image conics $\mathbf{A}_i, \mathbf{A}_j, \mathbf{A}_k$ alone, not computed from pixel intensities, gradients, or any appearance-derived quantity. Illumination therefore cannot influence the invariants except through the geometric accuracy of the fitted conics themselves.

These three invariants are proven to be algebraically independent and include all viewpoint-invariant information for a crater triad, generating a compact three-element descriptor that remains identical under arbitrary perspective projection.

**Projective invariant triad indexing and matching.** To address the critical balance between computational efficiency and accuracy, we structure the offline database as a highly optimized projective invariant triad index. Based on the Robbins crater catalog [35], each entry is first lifted into a 3D disk quadric $\mathbf{Q}_i^*$ via Eq. (1). Non-coplanar triads are systematically formed, and their three algebraically independent projective invariants $(J_i, J_j, J_k)$ are precomputed. To avoid the severe computational cost of matching during continuous descent, the descriptors are compressed into a 3-dimensional parameter space and indexed using a $k$-d tree. At runtime, as dynamic monocular images are captured, observed conics are grouped into triads, and their invariants are directly computed and queried via a nearest-neighbor search within the $k$-d tree. This indexing strategy ensures that the large-scale geometric representation can be searched efficiently without compromising the rigorous accuracy.

**Pose recovery.** Once crater correspondences are established, the absolute position of the camera is recovered by solving a linear system derived from the homographic relationship between each crater's local plane and the image plane. This formulation exploits the full rim contour of craters to estimate the camera position through the least-squares method, yielding a global measurement in the body-fixed frame for the state estimation filter.

### 3.4. Fusion of Relative and Absolute Navigation

The VO module provides high-rate direction-of-motion estimates but cannot recover the metric scale [19], while crater-based TRN provides absolute position fixes at a lower rate. An EKF fuses these two complementary sources to maintain a continuous, drift-corrected trajectory estimation.

The filter state consists of camera position and velocity in the lunar body-fixed frame, $\mathbf{x} = \begin{bmatrix} \mathbf{r}^\mathsf{T} & \mathbf{v}^\mathsf{T} \end{bmatrix} \in \mathbb{R}^6$ with camera attitude provided externally (*e.g.*, by a star tracker [21]) consistent with the known-attitude assumption of the pose recovery step. The state is propagated with a constant-velocity kinematic model, with process noise covariance derived from a piecewise-constant acceleration model, absorbing unmodeled dynamics such as gravity and thrust variations [17, 39].

**VO update.** At each frame, the VO module calculates a direction-of-motion $\tilde{\mathbf{s}}_k'$ with covariance $\mathbf{R}_{\mathbf{s}_k'}$. The measurement model normalizes the predicted velocity:

$$\mathbf{h}_{\text{VO}} = \mathbf{v}/\|\mathbf{v}\| \tag{4}$$

The corresponding Jacobian constrains direction only, leaving speed and position unobservable [19, 22].

**Crater-based TRN update.** When the crater-based TRN module successfully establishes correspondences, it provides an absolute position measurement $\tilde{\mathbf{r}}_k$ with $\mathbf{H}_{\text{absolute}}$ and measurement covariance $\mathbf{R}_{\text{absolute}} = \sigma_{\text{absolute}}^2 \mathbf{I}_3$. After a standard EKF measurement update, a hard correction is applied. The position state is set directly to $\tilde{\mathbf{r}}_k$ and the position covariance is reset to $\sigma_{\text{absolute}}^2 \mathbf{I}_3$ with the position–velocity cross-covariance zeroed. This reflects the fact that a catalog-based position fix carries no dependence on prior state history and can re-anchor the filter regardless of accumulated drift. A minimum frame interval between successive localization updates prevents absolute fixes from dominating the filter dynamics.

## 4. Dataset Construction under Controlled Illumination and Geometry

Evaluating the integrated pipeline across the descent conditions of interest requires a reproducible setting that spans illumination and viewing-geometry variability. We construct a simulation environment based on Lunar/LROC QuickMap [41], which aggregates pre-registered multiscale orbital data and enables synthesis of descent imagery under controlled geometric and photometric conditions without heavy local rendering. Compared to full 3D rendering engines [2, 26, 40], this avoids synthetic texture artifacts that affect appearance-sensitive algorithms [25].

The evaluation scenario targets the lunar south pole, a region with high scientific interest that presents extreme illumination variability due to low solar elevation angles. The descent trajectory is defined with 100 frames spanning altitudes from $100\,\text{km}$ down to $30\,\text{km}$, at latitude $-83.0°$, descends through the south pole (reaching $-89.997°$ near frame 54), and continues onto the opposite hemisphere to a final latitude of approximately $-87.1°$, traversing a total arc of roughly $10°$ across the south polar region. The $30\,\text{km}$ lower bound is set by the spatial resolution of the underlying LRO LOLA DEM (50 m/px). Below this altitude, the rendered images show spatial artifacts that compromise the evaluation of both feature matching and crater rim fitting. The virtual camera is configured with a $30°$ field of view (FOV) and $1024 \times 1024$ pixel resolution.

To systematically evaluate robustness to photometric and geometric variations, the dataset is organized into six scenario configurations (Fig. 3), each containing 100 frames along the same descent trajectory. Four illumination conditions under nadir-pointing view are sampled from QuickMap's pre-computed ephemeris layers, and two off-nadir viewing conditions with the illumination fixed at $(90°, 55°)$ introduce perspective distortion that stresses both VO feature matching and the projective invariant computation.

This parameterization yields 600 images in total (Tab. 1) and enables a controlled ablation that isolates illumination

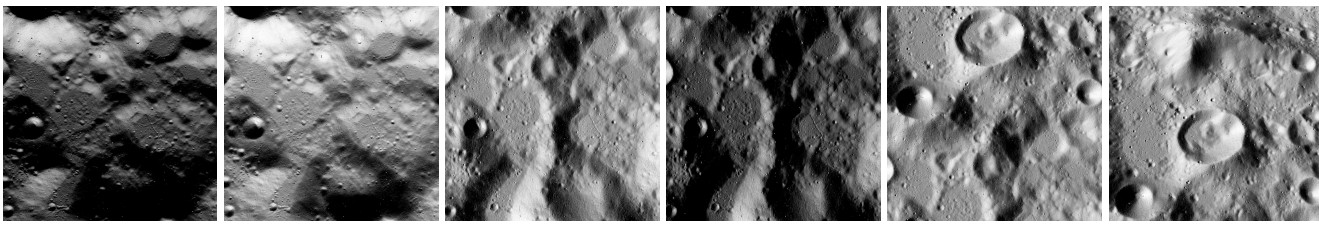

(a) S1: 0°, 55°, nadir    (b) S2: 0°, 75°, nadir    (c) S3: 90°, 55°, nadir    (d) S4: 90°, 75°, nadir    (e) S5: 90°, 55°, 15° off    (f) S6: 90°, 55°, 30° off

Figure 3. **Representative frames from the six QuickMap-based evaluation scenarios.** All scenarios share the same descent trajectory over the lunar south polar region and the same camera configuration (30° FOV, 1024 × 1024 px), but only solar geometry (azimuth, incidence) and camera tilt vary. S1–S4 cover the four nadir-pointing illumination conditions used to isolate photometric effects, where varying solar azimuth and incidence yield distinct shadow directions and contrast levels under the same viewing geometry. S5 and S6 introduce 15° and 30° off-nadir tilt under the fixed (azi 90°, inc 55°) illumination of S3, producing the perspective foreshortening that stresses the projective-invariant matching pipeline. All frames are sampled from a common altitude (100–30 km) in the descent sequence to enable direct visual comparison.

Table 1. **Dataset overview.** All scenarios share the same 100-frame descent trajectory (100–30 km altitude, south polar region) and camera configuration (30° FOV, 1024×1024 px). Solar geometry is parameterized by azimuth (azi) and incidence (inc) angles.

| Scenario | Illumination | Off-nadir | Frames |
|---|---|---|---|
| S1 | azi 0°, inc 55° | 0° | 100 |
| S2 | azi 0°, inc 75° | 0° | 100 |
| S3 | azi 90°, inc 55° | 0° | 100 |
| S4 | azi 90°, inc 75° | 0° | 100 |
| S5 | azi 90°, inc 55° | 15° | 100 |
| S6 | azi 90°, inc 55° | 30° | 100 |

Table 2. **Summary statistics.** Position error (km) and matching success rate of TRN baseline across scenarios ($\sigma_{\text{img}} = 0.5$ px, $p_{\text{FN}} = 0.316$, $p_{\text{FP}} = 0.430$).

| Scenario | VO-only | | TRN-only | | Integrated | | Match |
|---|---|---|---|---|---|---|---|
| | Mean | Final | Mean | Final | Mean | Final | Rate |
| S1 (0°, 55°) | 73.46 | 211.14 | 0.48 | 0.24 | 8.15 | 30.97 | 7% |
| S2 (0°, 75°) | 73.46 | 211.19 | 0.48 | 0.24 | 8.14 | 31.00 | 7% |
| S3 (90°, 55°) | 74.88 | 209.95 | 0.48 | 0.24 | 8.27 | 29.52 | 7% |
| S4 (90°, 75°) | 73.45 | 211.18 | 0.48 | 0.24 | 8.14 | 30.92 | 7% |
| S5 (15° off) | 73.35 | 211.62 | 0.66 | 0.20 | 26.16 | 90.90 | 4% |
| S6 (30° off) | 91.98 | 234.49 | N/A | N/A | 142.15 | 348.77 | 0% |

effects (across the four nadir scenarios) from viewpoint effects (across the three viewing angles with fixed illumination). All six scenarios share the same underlying descent trajectory, camera intrinsics, and crater catalog, ensuring that performance differences arise solely from the varied photometric or geometric conditions.

## 5. Experiments

### 5.1. Experimental Setup

**Baseline.** To isolate the contribution of each module, we evaluate three configurations: (i) *VO-only*, which propagates the state using only direction-of-motion estimates without absolute corrections; (ii) *TRN-only*, which returns crater-based localization fixes only at successful matches without state propagation between fixes; and (iii) *Integrated TRN*, which fuses both in the EKF.

**Synthesized crater observations.** The crater detection stage is bypassed and replaced with synthesized observations following [4]. Cataloged disk quadrics are projected into the image plane, retaining only those within the FOV. Three perturbations emulate upstream crater detection er-

rors: calibrated Gaussian noise ($\sigma_{\text{img}} = 0.5$ px as default) on the five ellipse parameters, random removal of in-FOV craters at rate $p_{\text{FN}}$ (false negative ratio), and injection of spurious ellipses at rate $p_{\text{FP}}$ (false positive ratio) relative to the number of true detections, with parameters sampled from the empirical in-FOV distribution. The rates $p_{\text{FN}} = 0.316$ and $p_{\text{FP}} = 0.430$ are anchored to the performance of the Mask R-CNN-based CDA of [36] ($F_1 = 63.1\%$), selected for its real-lunar validation on Chang'e-5 imagery, direct ellipse parameter output, and quantitative off-nadir evaluation. This protocol provides a controlled surrogate for upstream CDA performance while isolating the geometric matching and pose-recovery stages.

**Evaluation metrics.** The main evaluation criterion is the position error $\|\tilde{\boldsymbol{r}}_k - \boldsymbol{r}_k\|$ at each frame $k$, where $\tilde{\boldsymbol{r}}_k$ is the estimated position and $\boldsymbol{r}_k$ is the ground-truth position in the body-fixed frame. For VO-only and Integrated TRN, the velocity error $\|\tilde{\boldsymbol{v}}_k - \boldsymbol{v}_k\|$ is additionally reported.

### 5.2. Integrated Pipeline Evaluation

Tab. 2 summarizes the position error and matching success rate across all six scenarios at the default noise level ($\sigma_{\text{img}} = 0.5$ px), and Fig. 4 shows the three-way compari-

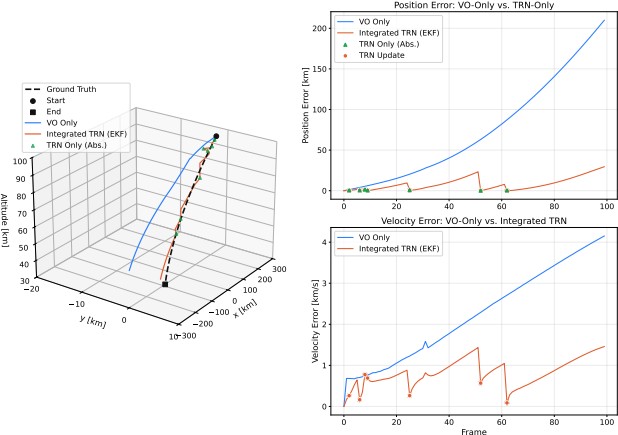

Figure 4. **Three-way comparison of VO-only, TRN-only, and integrated TRN on the S3 scenario.** Left: ground truth and estimated trajectories in the body-fixed frame. Right: position error (top) and velocity error (bottom) over frames. Orange dots indicate frames at which the crater-based localization module returns a verified position fix. TRN updates suppress accumulated VO drift up to frame 62. Beyond this point, insufficient crater visibility causes the pipeline to revert to VO-only propagation.

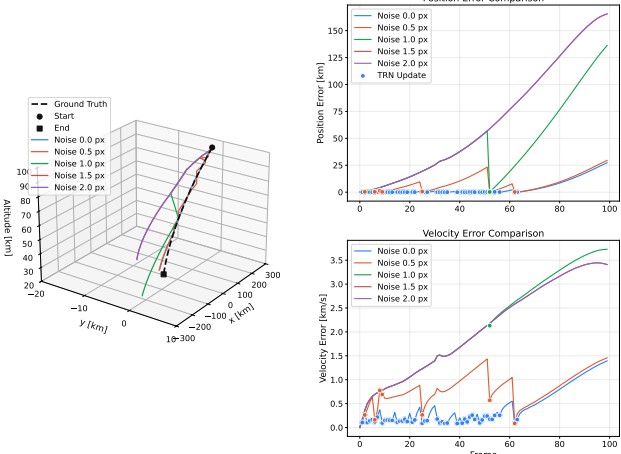

Figure 5. **Effect of observation noise on the integrated pipeline (S3 scenario).** Position error (top right) and velocity error (bottom right) are shown for $\sigma_{\mathrm{img}} \in \{0.0, 0.5, 1.0, 1.5, 2.0\}$ px. Colored dots on the position error plot mark successful TRN updates. At $\sigma_{\mathrm{img}} \leq 0.5$ px, TRN updates remain available through frame 62. At $\sigma_{\mathrm{img}} = 1.0$ px, matching becomes effectively rare, and at $\sigma_{\mathrm{img}} \geq 1.5$ px no valid matches are established.

son on S3. VO-only accumulates drift monotonically, with final position errors exceeding $200\,\mathrm{km}$ across all scenarios. This drift reflects the direction-only measurement model, which constrains velocity direction but provides no metric scale or absolute position information.

TRN-only, which evaluates the absolute position estimate from crater-based localization at each successful match, achieves a sub-km mean position error ($0.48\,\mathrm{km}$). This confirms that the projective invariant matching itself produces metric-grade fixes when correspondences are established. However, with a matching success rate of only 7%, TRN-only cannot maintain continuous state estimation between fixes, motivating the EKF fusion with VO for high-rate propagation.

The integrated pipeline reduces the mean position error by over 88% relative to VO-only ($8.27\,\mathrm{km}$ vs. $74.88\,\mathrm{km}$ on S3), demonstrating that sparse absolute fixes are sufficient to suppress accumulated drift when fused through the EKF. The position error profile (Fig. 4, orange dots) shows that error grows between TRN updates as the EKF relies on VO propagation, then is hard-reset at each successful localization.

The absence of TRN updates in the later portion of the trajectory reflects the decreasing number of cataloged craters within the camera field of view at lower altitudes. As the swath width of the camera decreases during descent, fewer crater triads are available for invariant computation, and no valid match can be established. Accordingly, the in-

tegrated pipeline reverts to VO-only propagation, and the final position error ($\approx 30\,\mathrm{km}$) reflects the drift accumulated over the remaining roughly 38 frames without correction.

## 5.3. Ablation Experiments

**Observation noise.** This ablation maps the first axis of the operational envelope. Fig. 5 shows the performance on S3 across $\sigma_{\mathrm{img}} \in \{0.0, 0.5, 1.0, 1.5, 2.0\}$ px. Final position errors increase non-linearly with noise: $27.34\,\mathrm{km}$ at $\sigma_{\mathrm{img}} = 0.0$ px, $29.52\,\mathrm{km}$ at $0.5$ px, and $136.24\,\mathrm{km}$ at $1.0$ px. At $\sigma_{\mathrm{img}} \geq 1.5$ px, the matching pipeline fails to produce any reliable fixes, and the final error converges to approximately $165.50\,\mathrm{km}$, which is a value determined by the VO-only drift accumulated over the entire trajectory. The upper bound of the noise axis under the realistic detection rates thus lies near $\sigma_{\mathrm{img}} \approx 0.5$ px.

The triad invariants are computed from the fitted image conics via Eq. (3), so any perturbation of the conic parameters propagates directly into the invariants. During the invariants calculation, the eigendecomposition of the degenerate conic pencil amplifies the error of the input conics. The same mechanism explains the catastrophic failure at $30°$ off-nadir discussed below. Large off-nadir angles yield substantial perspective distortion of the projected ellipses, which acts as an effective increase in the geometric noise on the conic parameters, causing the invariants to degrade beyond the matching threshold.

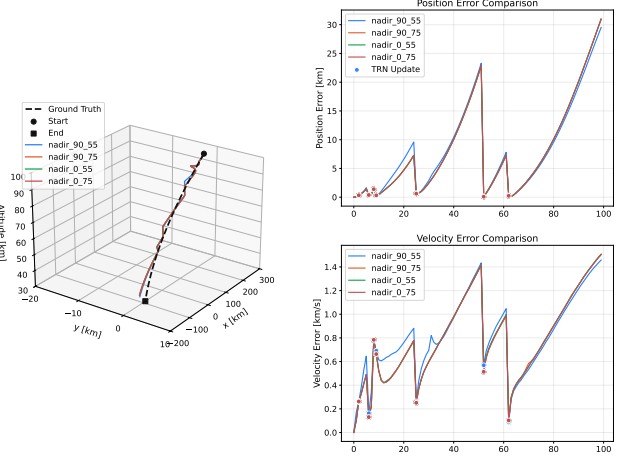

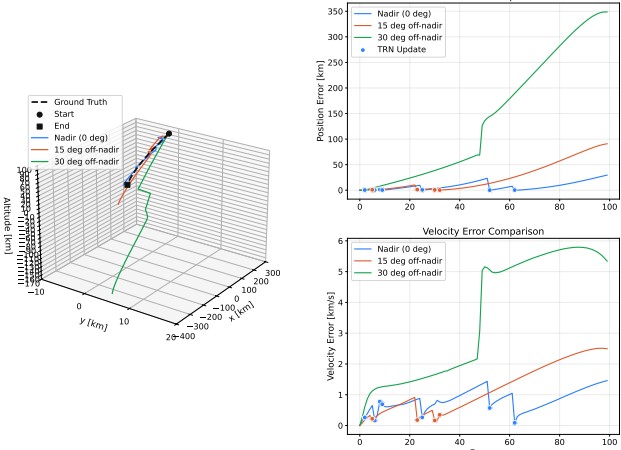

**Figure 6. Integrated TRN performance across four illumination conditions under nadir-pointing view.** The four scenarios (S1–S4) result in nearly identical position/velocity error profiles.

**Figure 7. Effect of off-nadir viewing angle on the integrated pipeline.** At 15° off-nadir, TRN updates decrease from 7 to 4 successful fixes, with the last fix occurring at frame 32. At 30° off-nadir, no successful match is established (0% matching rate), and the final error grows to 348.77 km, exceeding the VO-only baseline.

**Illumination.** Across the four nadir-pointing scenarios (S1–S4), the integrated TRN pipeline produces nearly identical error profiles (Fig. 6), with mean position errors within 8.14–8.27 km and matching success rates uniformly at 7%. This result confirms the prediction that the triad invariants are functions of the fitted image conics alone, so illumination can enter the pipeline only through the geometric accuracy of those conics, not through the invariant calculation itself. The illumination axis of the operational envelope is therefore unbounded at the representation level. Under the synthesized observation protocol, the four scenarios differ only in VO matching, whose effect on the integrated error is small relative to TRN updates. A full pipeline evaluation including appearance-based crater detection remains future work.

**Viewing geometry.** Fig. 7 compares nadir-pointing, 15°, and 30° off-nadir viewing at fixed illumination (azimuth 90°, incidence 55°). The matching success rate drops from 7% to 4% to 0% across the three configurations. The viewing-geometry bound of the envelope therefore lies between 15° and 30° off-nadir.

Two observations characterize the behavior within and beyond this bound. Within the envelope, despite only four successful matches, the 15° configuration achieves a final position error of 90.90 km, remaining bounded within the same order of magnitude as the nadir result (29.52 km). This indicates that the EKF can tolerate substantial reduction in TRN update frequency, provided a few fixes are distributed across the trajectory to periodically re-anchor the position state. Beyond the envelope, at 30° off-nadir, the

complete absence of successful matches eliminates all absolute corrections and produces a final error of 348.77 km, which exceeds the VO-only baseline (234.49 km). The reversal reflects that the EKF constrains speed only indirectly through TRN updates. Due to their absence, the speed estimate drifts without correction.

## 6. Conclusion

We construct an offline projective invariant triad index by lifting geo-referenced crater catalog parameters into disk quadric primitives [4, 35]. We use the crater-based TRN coupled with monocular VO and an EKF as the instrument through which the geometric properties of this representation become observable as trajectory-level navigation performance. Our experiments characterize the operational envelope of the representation along three axes. We establish that illumination invariance is unbounded at the representation level. The observation noise axis is bounded at $\sigma_{img} \approx 0.5$ px, and the viewing geometry axis is bounded between 15° and 30° off-nadir. Both bounds reflect the same underlying mechanism: geometric perturbations are amplified through the eigendecomposition of the degenerate conic pencil. Within the envelope, sparse absolute fixes available at 7% of frames suffice to suppress accumulated VO drift by over 88%. Future work will extend this characterization to additional descent trajectories with diverse landing sites, and to a full pipeline that includes appearance-based crater detection.

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
