# OpenReview forum: "Projective Invariant Triad Indexing for Integrated Lunar Terrain Relative Navigation"
_thecvf.com/CVPR/2026/Workshop/3D4S — CVPR 2026 Workshop 3D4S Poster_

### Official Review · Reviewer_dCPX · 2026-04-19
**Relevant planetary-science application, but limited novelty and an over controlled evaluation**

**Rating:** 5
**Confidence:** 4

**Review:**

This is an application-oriented paper with a clear problem motivation, a complete engineering pipeline, and generally smooth/clear writing.  The direction is reasonable and relevant to planetary science and scientific 3D geometry applications. However, as a research paper, its main weaknesses are limited methodological novelty and over-simplified/idealized experimental setup. At its current stage, I see it more as a useful system integration and preliminary simulation study than as a mature workshop paper.

For the strengths of this paper, the research problem is important and well-motivated. The proposed geometric idea is also reasonable and I also appreciate the honesty about failure cases. Specifically, it not only just reports successful results but also shows that the method fails under high observation noise and at a 30-degree off-nadir viewing angle. This kind of operational envelope analysis is helpful. Moreover, the system integration has good practical value.

However, I have a few concerns: The most important issue is limited novelty. Many of the paper’s core mathematical tools, including disk quadrics, crater triads, projective invariants, and indexed search over crater combinations, come from prior crater identification literature. The main new contribution appears to be connecting existing projective-invariant crater matching to a VO/EKF pipeline and evaluating it over a continuous descent trajectory. In this case, it seems closer to system integration than to a clearly new method. The second issue I would flag is the experiments bypass the hardest part of the problem: crater detection. The authors do not actually detect craters from images. Instead, they project ideal craters from the catalog and add Gaussian noise. This makes the experimental setting a bit too clean.  Because of this, I would be cautious about the paper’s claim of illumination robustness. A more accurate statement would be: if accurate crater geometry observations are already available, then the projective invariants themselves do not depend on illumination. This is not the same as showing that the full system is robust to illumination changes. In addition, the paper has an inverse crime risk: the simulation setup is quite close to the assumptions used by the algorithm. The observations are generated from a catalog, and the matching is also performed using catalog-based geometry. Adding Gaussian noise does not fully capture real-world catalog mismatch, DEM mismatch, crater rim degradation etc. Lastly, the statistical evidence is insufficient as experiments use only one trajectory. There are no multiple random seeds, no multiple descent profiles, no confidence intervals and etc. For a navigation system, this is not enough. The reported result that the mean position error is reduced by more than 95% may depend strongly on the chosen trajectory, crater density, update timing, and simulation assumptions.

---

### Official Review · Reviewer_PL3t · 2026-04-25
**Projective Invariant Triad Indexing for Integrated Lunar Terrain Relative Navigation**

**Rating:** 7
**Confidence:** 3

**Review:**

This paper presents a projective invariant crater-triad indexing method for lunar terrain relative navigation and integrates it with monocular visual odometry through an EKF. The problem is well motivated: VO provides continuous relative motion estimates but suffers from scale ambiguity and accumulated drift, while crater-based TRN can provide sparse absolute position fixes. The proposed integration is logical and practically relevant for lunar descent navigation.

A key strength of the paper is its use of 3D crater geometry and projective invariants to bridge the gap between offline orbital crater catalogs and runtime descent imagery. The simulation study over the lunar south pole is also useful, especially the analysis across observation noise, illumination, and off-nadir viewing angles. The results show that sparse crater-based fixes can significantly reduce VO drift within the tested operational envelope.

However, the evaluation is not fully end-to-end. The crater detection stage is bypassed, and crater observations are synthesized, which weakens the claim of illumination robustness in realistic settings. The comparison is also mainly against VO-only and TRN-only configurations, rather than alternative crater matching or indexing methods. In addition, the method fails at 30° off-nadir view, and the paper could discuss this limitation more deeply.

Overall, this is a solid workshop paper with a clear motivation, coherent system design, and promising results. The work would be stronger with an end-to-end crater detection pipeline, more trajectories, and comparisons against other crater-matching baselines.

---

### Official Review · Reviewer_JuW5 · 2026-04-25
**Projective Invariant Triad Indexing for Lunar TRN: A Well-Motivated but Experimentally Limited Navigation Framework**

**Rating:** 5
**Confidence:** 4

**Review:**

## Strengths

1. Important and practical problem.
   Lunar landing navigation is a safety-critical application where GNSS is unavailable or unreliable. The paper clearly motivates why VO alone is insufficient and why sparse absolute TRN updates are needed.

2. Clear geometric formulation.
   The paper gives a clean explanation of how 2D crater catalog ellipses are lifted into 3D disk quadrics and how triad projective invariants are computed. The use of projective invariants is suitable for bridging heterogeneous orbital catalog data and monocular descent imagery.

3. Good system-level integration.
   Instead of only studying crater matching in isolation, the authors integrate crater-based TRN with VO using an EKF. This makes the evaluation more meaningful from a navigation perspective.

## Weaknesses / Discussion

1. Novelty is limited.
   The core projective-invariant crater triad formulation is heavily based on prior work, especially Christian et al. The paper’s main novelty is applying this representation in an integrated VO+EKF pipeline and evaluating its operational envelope. This is useful engineering, but the theoretical or algorithmic novelty is moderate rather than strong.

2. The crater detection stage is bypassed.
   A major limitation is that the paper does not evaluate a full end-to-end TRN pipeline. The crater detection algorithm is bypassed, and observed crater conics are synthesized by projecting cataloged disk quadrics and injecting Gaussian noise. This isolates the matching module, but it also makes the evaluation much easier than real descent imagery. In practice, illumination changes, shadows, partial crater visibility, terrain texture, and detection false positives/false negatives are central challenges. Therefore, the claim of illumination robustness is only valid at the invariant representation level, not at the full system level.

3. Dataset and simulation scale are limited.
   The evaluation uses one lunar south pole descent trajectory with six scenario configurations and 600 total images. This is not enough to convincingly demonstrate general robustness across lunar landing conditions. The paper would be stronger with multiple descent trajectories, different landing regions, different crater densities, different altitude ranges, and more diverse camera configurations.

---

### Decision · Program_Chairs · 2026-04-28

Accept (Poster)